



# A hydrological framework for persistent river pools in semi-arid environments

Sarah A. Bourke[1], Margaret Shanafield[2], Paul Hedley[3], Shawan Dogramaci[3]

[1]School of Earth Sciences, University of Western Australia, Crawley WA 6009 Australia
[2]College of Science and Engineering, Flinders University, Bedford Park, SA 5042, Australia
[3]Rio Tinto Iron Ore, Perth, WA 6000 Australia

*Correspondence to*: Sarah A. Bourke (sarah.bourke@uwa.edu.au)

**Abstract**

Persistent surface water pools along non-perennial rivers represent an important water resource for the plants, animals, and humans that inhabit semi-arid regions. While ecological studies of these features are not uncommon, these are rarely accompanied by a rigorous examination of the hydrological and hydrogeological characteristics that create or support the pools. Here we present an overarching framework for understanding the hydrology of persistent pools based on data from 22 pools in the Hamersley Basin in Western Australia. Three dominant mechanisms that control the occurrence of persistent pools have been identified; perched pools, through flow pools and groundwater discharge pools. Groundwater discharge pools are further categorized into those that are present because of a geological contact or barrier, and those that are controlled by topography. A suite of diagnostic tools (including geological mapping, hydraulic data and hydrochemical surveys) is generally required to identify the mechanism supporting persistent pools. Perched pools are sensitive to climate variability but their persistence is largely independent of groundwater withdrawals. Water fluxes to pools from alluvial and bedrock aquifers can vary seasonally and resolving these inputs is generally non-trivial. The susceptibility of through-flow and groundwater discharge pools to climate variations and groundwater withdrawals depends on the mechanism of pool persistence and the spatial distribution of stressors relative to the pool. Although this framework was developed in the context of north-western Australia, this framework can also be applied to pools and springs found along non-perennial rivers around the world.

## 1 Introduction

Permanent or almost permanent water features along non-perennial rivers (hereafter referred to as "persistent pools") represent an important water resource for the plants, animals, and humans that inhabit arid regions. These features typically hold residual water from periodic surface flows, but also may receive input from underlying aquifers, and have alternately been termed pools (Bogan and Lytle, 2011; Jaeger and Olden, 2011; John, 1964), springs (Cushing and Wolf, 1984), waterholes (Arthington et al., 2005; Bunn et al., 2006; Davis et al., 2002; Hamilton et al., 2005; Knighton and Nanson, 2000; Rayner et al., 2009), and wetlands (Ashley et al., 2002). They exist in a wide variety of habitats, from the isolated wetland in Kenya (Ashley et al., 2002) to the arid southwest of the U.S. (Bogan and Lytle, 2011) and all across Australia (Arthington et al., 2005; Bunn et al., 2006; Davis et al., 2002). Several studies have confirmed that these water features are highly diverse (Shepard, 1993) and can vary significantly in water quality (Stanley et al., 1997). Persistent pools are also often of cultural significance



(Finn and Jackson, 2011; Yu, 2000) and are sensitive to changing climate and human activities (Bunn et al., 2006; Jaeger and Olden, 2011). However, their hydrology is typically only poorly understood.

In semi-arid and arid regions, these pools are often situated within non-perennial stream channels. The intermittent/ephemeral streams that host these pools are receiving renewed attention (Acuña et al., 2014; Datry et al., 2016). Concurrently, pools within these stream systems have received attention for the role they play as a seasonal refuge (Goodrich et al., 2018), and with regards to connectivity between riparian ecosystems (Godsey and Kirchner, 2014). By far, the published literature on these features focuses on the ecological processes and patterns. For example, they have been shown to host unique fish assemblages (Arthington et al., 2005; Labbe and Fausch, 2000), macroinvertebrate communities (Bogan and Lytle, 2011), and primary productivity (Cushing and Wolf, 1984). Recently, it was shown that the structure, but not composition, of these pools mirrors that of perennial rivers (Kelso and Entrekin, 2018).

However, rarely are these ecological studies accompanied by a rigorous examination of the hydrological and hydrogeological characteristics that provide a setting for these ecologic communities. There are isolated studies that examine the composition of water within persistent pools. For example, Hamilton et al. (2005) examined the role of evaporation in controlling pool persistence along a large dryland river in south central Australia using major ion chemistry and stable isotopes. Fellman et al., (2011) used $\delta^2H$ and $\delta^{18}O$ values of pool water, rainfall, and groundwater combined with pool water measurements of C, N, and P and dissolved organic matter (DOM) fluorescence characteristics to determine the origin of selected surface water pools of a dryland river in northern Western Australia. However, more typically the studies of these pools only describe the seasonal persistence of flow and basic hydrologic parameters (typically temperature and salinity, sometimes also oxygen) relevant to the species under investigation. There are also studies that examine seasonal variability of habitat conditions for certain species in intermittent stream pools (Cushing and Wolf, 1984;Jaeger and Olden, 2011;John, 1964) but do not relate individual pools to the larger geologic or hydromorphic landscape phenomena that led to their occurrence. Thus, an overarching hydro-geomorphological framework to describe how and under what conditions these pools persist is still lacking.

Here we present an overarching framework for understanding the hydrology of persistent pools based on data from 22 pools in the Hammersley Basin in Western Australia. This region is characterized by relatively low population with little agricultural development but large volumes of resource extraction, especially iron ore mining. Persistent pools, which are common across this landscape, act as ecological refugia during dry periods, and have significant cultural value to Aboriginal communities and Traditional Owners. Because of this intersection of economic, ecological and cultural value, substantial investments have been made in the collection of geographical, hydrological and geological data across the basin to understand persistent pools in this region. We use this data to develop a framework that identifies four key mechanisms for supporting the occurrence of persistent pools. We then critique the tools available for identifying the mechanism controlling pool persistence and discuss the susceptibility of each pool type to shifts in climate or groundwater abstraction.

## 2 Study Site Description

The Hamersley Basin in north-west Australia (Figure 1) has an arid-tropical climate with a wet season from October to April and a dry season from May to September (Sturman and Tapper, 1996). Average annual rainfall



is less than 300 mm yr$^{-1}$ with most rain falling between December and April (www.bom.gov.au). Annual rainfall statistics can vary dramatically, depending on the influence of thunderstorms and cyclone activity. Thunderstorm activity is commonly highly localised, limiting the potential for spatial interpolation of data from individual monitoring sites. Annual evaporation is around 3000 mm yr$^{-1}$ (www.bom.gov.au), or about ten times annual rainfall, so that permanent surface water is rare.

Ranges, spurs, and hills are separated by broad alluvial valleys with numerous deep gorges created by differential erosion. During large flood events, runoff creates sheet flow along the main channel and the extensive floodplain can remain flooded for several weeks. In the absence of cyclonic rainfall, surface water is generally limited to a series of disconnected pools along the main channels. The valleys are filled with up to 100 m of consolidated and

unconsolidated Tertiary detrital material consisting of clays, gravels, and chemical precipitates. Quaternary sediments along the creek-lines and incised channels (incised on the order of m's) consist of coarse, poorly sorted gravel and cobbles. Fresh groundwater is abundant throughout the region, both within the Archean basement rocks, where permeability is increased via weathering, fracturing or mineralisation, and within the Tertiary and Quaternary sediments.

### 3 Mechanisms controlling the occurrence of persistent pools

Four dominant mechanisms that control the occurrence of persistent pools have been identified using data from across the Hammersley Basin (Table 1). Persistent pools can be a combination of more than one of these mechanisms, but there is commonly one dominant mechanism that ultimately controls the location and persistence of the pool. For example, a pool may contain a mixture of water from streambed sediments and regional


groundwater during certain hydroperiods, but the pool wouldn't persist in that location without groundwater discharge from the regional aquifer.

#### 3.1 Perched pools

Perched pools are topographic lows that are disconnected from the groundwater system but retain rainfall and runoff during the dry season (Fig. 2). The persistence of these pools depends on a) shading from direct sunlight


and/or, b) sufficient water volume that it is not completely depleted by evapotranspiration during the dry season. These types of pools are characterized by hydraulic gradients between the water level in the pool and local groundwater that indicate perched conditions. Such perching is only possible in the presence of a low-conductivity layer between the pool and the water table, which causes disconnection between streams and underlying aquifers (Brunner et al., 2009). In the Hammersley Basin, these are more commonly found in elevated, hard-rock areas


where erosion has created a deep pool that is shaded to minimise evaporation. Less commonly, these types of disconnected pools can also form in the lower catchment where clay sediments have been deposited over the alluvium.

#### 3.2 Through-flow pools


Through-flow pools are surface expressions of the water table in streambed sediments that are sustained by a direct hydraulic connection between the pool and the water table within the streambed (Fig. 3). Pools supported by this mechanism will persist provided the reservoir of water stored within the streambed sediments is larger





than losses to evaporation during the dry period. The water level in these pools is effectively a window into the water table within the streambed sediments. Once the pool is isolated from the flowing stream, the water flow through these pools is effectively that of an elongated, through-flow lake with inflow from the subsurface at the top of the pool and outflow to the subsurface at the bottom of the pool (Townley and Trefry, 2000). The hydraulic

gradient in these pools can change seasonally in response to recharge by rainfall events and subsequent depletion of water stored in the sediments; a process analogous to "bank storage" adjacent to flowing streams. Through-flow pools are commonly found along contemporary drainage lines in the Hammersley basin, numbering on the order of 10's of pools along a major drainage line 100's of kms long.

### 3.3 Groundwater discharge pools

### 3.3.1 Geological contacts and barriers to flow

Geological contacts and barriers to flow that force outflow of groundwater to the surface can cause pools to occur, either where groundwater flow through a catchment is forced through a narrow constriction by hard-rock ridges (Fig. 4a), or a low-permeability layer in the subsurface forces groundwater to flow to the surface (Fig. 4b). Pools supported by groundwater discharge over a low-permeability layer operate similarly to previously described

contacts springs (Kresic and Stevanovic, 2010; Bryan, 1919), but as far as we are aware the discharge of groundwater through a narrow constriction as a mechanism for surface water generation has not previously been described in published literature. The geological barriers controlling these pools result in lateral hydraulic gradients across the pool feature. The persistence of these pools will be a function of the reservoir of water that they drain. If the constrained catchment is large, they may persist through multiple dry-seasons; if the constrained

catchment is small, they may not persist through more than one dry-season. These types of pools are common in the Hammersley Basin where the intersection of fluvial deposits and erosion-resistant, low permeability basement rock is common.

### 3.3.2 Topographically controlled seepage from regional aquifers

Pool persistence can be sustained by groundwater seepage from regional aquifers in the absence of geological

barriers or contacts if there is a topographic low that intersects the regional water table. These types of pools generally occur where differential erosion causes a difference in topography. To the best of our knowledge, in the Hammersley Basin, this mechanism is restricted to pools supported by unconfined aquifers (Fig. 5), which is similar to depression springs (Kresic and Stevanovic, 2010; Bryan, 1919). Pools may also be sustained by topographically controlled seepage from confined aquifers if there is a fault or fissure that acts as a conduit to

groundwater flow (different to Fig. 4a because there is no geological transition to sustain a hydraulic gradient across the pool). Because the locations of these pools will be regional groundwater discharge zones, local recharge to the source aquifer from the pool should be negligible. Groundwater discharge along fractures or faults has been identified as an important mechanism for groundwater discharge to the Fitzroy River in northern Australia (Harrington et al., 2013) but the significance of this regional groundwater discharge to individual persistent pools

is not yet known. Topographically controlled discharge from a confined aquifer is analogous to artesian mound springs like those found in the Great Artesian Basin of central Australia (Ponder, 1986).



## 4 Diagnostic tools for elucidating hydraulic mechanisms supporting pool persistence

### 4.1 Landscape position and remote sensing

Landscape position can provide some clues as to the mechanism controlling the persistence of a given pool (Fig. 6). For example, a pool located high in the catchment on impermeable basement rock is likely to be a perched pool. A pool that is immediately prior to a ridge that constrains the catchment is likely to be supported by geologically constrained groundwater discharge. The presence of active deposition of geological precipitates can also be indicative of pool mode of occurrence with carbonates associated with groundwater discharge and subsequent degassing of $CO_2$ (Mather et al., 2019). Precipitation of evaporative salts is less definitive. In the Hammersley Basin, any salts deposited at perched pools tend to be washed away during flood events, so that salt accumulation generally indicates hydraulic connection between the pool and underlying aquifer. Mapping the persistence of vegetation and water in the landscape based on remotely sensed data (i.e. NDVI or NDWI) can be used to identify pools that persist (Haas et al., 2009; Soti et al., 2009; Alaibakhsh et al., 2017), but this alone does not explain the hydraulic mechanism determining the location of the pool. Combining these vegetation indices with aerial geophysics (i.e. AEM) can aid in developing a better understanding of hydraulic mechanisms in remote areas, allowing the identification of low-permeability layers or geological structures that are not obvious from aerial photographs.

### 4.2 Hydrography and pool water balances

Direct measurement of water balances in arid and semi-arid regions can be logistically difficult. Rainfall (and therefore runoff) in arid and semi-arid environments is commonly patchy and water fluxes can be either too large to measure (streamflow during a cyclone) or too small to measure directly (dry-season groundwater seepage fluxes). Because these pools are commonly areas of environmental and cultural significance, it is essential that appropriate approvals and permissions are obtained prior to the installation of monitoring infrastructure, and this may restrict the types of data that can be collected.

Once a pool becomes isolated from the flowing river, and in the absence of rainfall, a general pool water balance is given by;

$$\frac{\partial V}{\partial t} = Q_i - Q_o - EA \tag{1}$$

where $V$ is the volume of water in the pool ($L^3$), $t$ is time (T), $Q_i$ is the water flux from the subsurface into the pool ($L^3T^{-1}$), $Q_o$ is the water flux out of the pool into the subsurface ($L^3T^{-1}$), $E$ is the evaporation rate (L $T^{-1}$) and $A$ is the surface area of the pool ($L^2$). Modified versions of this general water balance can be defined for each pool type (Table 1). For perched pools, which are disconnected from the groundwater system, $Q_i=Q_o = 0$, so that the only component of the water balance is water loss through evaporation. Through-flow pools are connected to the water stored in the streambed alluvium with influx and efflux rates that can change over time. For groundwater discharge pools, influx will dominate over efflux ($Q_i > Q_o$) and the influx can be from the streambed sediments or the adjacent aquifer. If the groundwater discharge is over an impermeable aquiclude (see Fig. 3b) there will commonly be a seepage zone up-gradient of the pool so that water influx is via surface inflow, but outflow is to the subsurface and forms a source of groundwater recharge. If the groundwater discharge is controlled by topography, the pools are sites of regional groundwater discharge so that $Q_o$ should be negligible.



The water level in the pool, $h_p$ (L), can be routinely measured by installing pressure transducers, but conversion of water levels to pool water volume requires knowledge of pool bathymetry, and the relationship between $h_p$, $V$ and $A$ will change over during the dry season as the pool water level recedes and also annually if pool bathymetry is altered by flood events. Evaporation rates can be taken from regional data or empirical equations, but actual

losses can vary depending on solar shading, wind exposure and transpiration. If groundwater levels in the source aquifer are known, then $Q_i$ (or $Q_o$) can be estimated from Darcy's Law;

$$Q_i = K \frac{\Delta h}{\Delta x} A_i \tag{2}$$

where $K$ is hydraulic conductivity, $\frac{\Delta h}{\Delta x}$ is the hydraulic gradient between the pool and the source aquifer, and $A_i$ is

the area over which the groundwater inflow occurs (which will usually be less than the total area of the base of the pool). The major limitations of this approach are that $K$ of natural sediments varies by orders of magnitude, and that the area of groundwater inflow needs to be assumed or estimated using a secondary method. Hydraulic gradients between pools and streambed sediments can be measured using monitoring wells or temporary drive points, with $\Delta h$ usually on the order of centimetres at most. Determination of the hydraulic gradient between

regional aquifers requires that the water level in the pool has been surveyed to a common datum and there is a monitoring well near the pool to measure the groundwater level relative to that datum. For pools with visible surface inflow or outflow, it may be possible to measure these rates using flow gauging (or dilution gauging), but the relatively small flow rates and bifurcation of the flow can make this challenging. In shallow, groundwater dominated lakes, geophysical methods have also been used to determine local gradients (and therefore whether

flow-through conditions occur) (Ong et al., 2010; Befus et al., 2012); such methods would likely also be appropriate for pools.

### 4.3 Hydrochemistry and pool mass balance

Numerous studies of streams and lakes have employed hydrochemical and mass balance approaches to quantify water sources (Cook, 2013; Sharma and Kansal, 2013). Some of these methods are also applicable in persistent

pools, but may require modification, or an iterative approach that allows for refinement of the methods as the mechanism supporting the pool is elucidated.

Dissolved ions (including EC) and stable isotopes are relatively cheap and easy to measure, but their application to identify or quantify water sources in the Hamersley Basin is commonly limited by overlapping values (Bourke et al., 2015). Time series of EC and stable isotopes can be used to indicate relative rates of evaporation and

through-flow (Siebers et al., 2016; Fellman et al., 2011). In the case of EC, transient changes in response to flood-recession cycles can indicate the mechanism supporting the pool, with a flush of high EC water immediately following floods, followed by re-equilibration commonly measured in pools controlled by geologically-driven groundwater discharge (Fig. 7). This would contrast with perched pools, which would not continue to evapo-concentrate until the next flood event. Similarly, temporal variation, or consistency of stable isotope values can

provide valuable insights into the mechanisms supporting pool persistence. Groundwater seepage from a regional aquifer will have a relatively consistent isotopic value, while a pool isolated from the groundwater source will experience isotopic enrichment through evaporation (Fig. 8).





Radon-222 is a commonly applied tracer in studies of surface water – groundwater interaction, and [222]Rn mass balances have been effective for identifying groundwater contributions to streams and lakes (Cook, 2013; Cook et al., 2008). The application of this approach in persistent pools is limited by the fact that these pools tend to be poorly mixed with substantial spatial variability in [222]Rn activity along the pools (Fig. 9). Other groundwater age

indicators have been applied in streams to identify groundwater sources, but their applicability in pools is yet to be determined. Given that shallow stagnant-ish water is common, [14]C or [3]H which don't rapidly equilibrate with the atmosphere (Bourke et al., 2014; Cook and Dogramaci, 2019) are likely to be better than isotopic tracers (e.g. [4]He) that equilibrate rapidly (Gardner et al., 2011). If a mass balance approach is applied, then hydraulic measurements to constrain the pool water balance should be made in conjunction with hydrochemical sampling

to ensure that the water balance is appropriately reflected in the mass balance.

Temperature measurements have been used extensively to identify and quantify water fluxes across streambeds and lakebeds (e.g. Shanafield et al.,2010; Lautz, 2012; Rau et al., 2014). In persistent pools, temperatures at the water sediment interface can be used to map zones of groundwater inflow (Conant, 2004). In arid zones groundwater temperatures will often be warmer than pool temperatures and this type of survey is best conducted

at dawn when the temperature gradient between pool and groundwater is at a maximum and there are no confounding effects from direct solar radiation. This mapping can be conducted using point sensors or thermal cameras, but in natural water bodies this method has primarily found success at thermal springs where the temperature difference between surface waters and groundwater inflows is large (Briggs et al., 2016; Cardenas et al., 2011). Vertical profiles of temperature can also be used to estimate vertical fluid fluxes, but the application of

this approach in pools with overly course alluvial sediments (commonly through-flow pools) is likely to be limited by lateral flow within the subsurface when $K_h > K_v$ (Rau et al., 2010; Lautz, 2010). Analytical solutions for temperature-based flux estimates also break-down at low flux rates where the difference between convection and conduction is difficult to determine (Stallman, 1965). Recently developed instrumentation for measuring 3D flux fields (Banks et al., 2018) shows promise, but installation in course alluvial sediments like those commonly found

in arid streambeds remains a challenge. Point-scale measurements also require up-scaling and these methods may not be applicable in fractured hard-rock pools.

## 5 Susceptibility of persistent pools to changing hydrological regimes

Robust water resource management in semi-arid regions requires an understanding of the ways in which human activities or shifting climates can alter pool water balances and/or the duration of persistence. Here we present

general guidance on the susceptibility of pool types to changes in rainfall and groundwater abstraction (Table 1). Intuitively, the size of the reservoir (surface catchment or groundwater storage) that supplies water to the pool seems like it would be a key factor in determining the susceptibility of persistent pools to changing hydrological regimes. However, in the case of pools reliant on surface catchments in particular (perched or through-flow pools), the patchiness of rainfall and substantial transmission losses mean that catchment size alone is unlikely to be a

robust predictor of resilience. For perched and through-flow pools, susceptibility to changes in rainfall, or reductions in catchment size due to mining, are likely to be more dependent on proximity to the pool, rather than simply the volume of the reduction. In arid systems that are storage-limited (like the Hamersley Basin), increasing cyclonic activity and heavy rainfall events my not necessarily result in increased pool persistence, particularly in pools closest to the location of rainfall. If subsurface storage up-gradient of the pool is already filling during the





wet season, subsequent rainfall will increase streamflow downstream, but not result in increased subsurface storage in the reservoir supporting the pool.

For pools sourced from groundwater reservoirs, head gradients (and groundwater discharge rates) supporting the pools may be small (Δh on the order of cms), so that small changes in hydraulic head can potentially have a

detrimental impact on the pool and cause the pool to dry out (particularly for topographically controlled pools). For through-flow pools, the water balance (Table 1) is dominated by water outflow from contemporary fluvial deposits but abstraction from regional groundwater could impact the pool if there is hydraulic connection. The volume of groundwater storage in the source reservoir can indicate the resilience of pools to hydrological change (i.e. a longer groundwater system response time), but impacts will also depend on the distance from the recharge

zone or groundwater abstraction (Cook et al., 2003). The time-lag prior to a decrease in groundwater outflow to the pool, and shape of the response (i.e., is it a slow decline, or sharp decrease), will also depend on the spatial distribution of the forcing (i.e. distance from recharge or groundwater abstraction) (Cook et al., 2003; Manga, 1999). Thus, focussed groundwater abstraction close to a pool will cause a larger and faster reduction in groundwater outflow than diffuse abstraction across the aquifer, or abstraction further away (Cook et al., 2003;

Theis, 1940). For example, groundwater is pumped from within 1 km of a pool will result in a rapid decrease in discharge (months to years). On the other hand, the same volume of abstraction distributed throughout the catchment will result in a more gradual decline in groundwater discharge to the pool (years to decades). Susceptibility can be further modified by geological barriers, which may not be obvious from the surface topography or regional geological maps (Bense et al., 2013), but can isolate pools from the regional groundwater

system and either i) increase susceptibility to pumping within the connected aquifer, or ii) reduce susceptibility if the pumping is on the other side of the barrier.

## 6 Discussion

Here we present a hydrogeological framework for four types of pools commonly found along non-perennial rivers.

This framework establishes the conceptual models and nomenclature required for a more rigorous approach to the study of persistent pools. To date, the treatment of the hydrology of persistent river pools in published literature has been largely descriptive, vague or tangential to the main theme of the paper (Thoms and Sheldon, 2000). For pools receiving groundwater discharge, the hydraulic mechanism supporting pool persistence can be similar to a spring. It has now been 100 years since groundwater springs were documented in published literature (Bryan,

1919; Meinzer, 1927; Meinzer, 1923) and while frameworks exist there is not one definitive classification system. Papers published more recently have not yet applied this earlier work to our understanding of persistent river pools and are either largely descriptive, or focus on the context (karst vs desert) of spring water quality and do not encompass the mechanisms we have observed that support persistent river pools (Springer and Stevens, 2009; Shepard, 1993; Alfaro and Wallace, 1994). Thus, while the existing literature hints at the hydrologic and geologic

constraints imperative to pool persistence, the framework presented here provides a more scientific characterisation as required to sufficiently understand and protect persistent pools in arid regions.

Although this framework was developed in context of the north-western Australia, this framework can also be applied to pools and springs found along non-perennial rivers around the world. Within Australia, these types of residual pools are found throughout the country. For example, along Cooper Creek in central Australia,



geochemical and isotopic studies revealed a lack of connection to groundwater, that convergence of flows at the surface and subsequent evaporative water loss controlled water volumes in many pools (Knighton and Nanson, 1994; Hamilton et al., 2005). These pools, situated in depressions caused by erosion through sandy subsurface layers, are a good example of perched pools (low-conductivity layer for perching not elucidated). We have

distinguished between geological or topographic control on groundwater discharge, but this distinction may not be critical from management perspective. For example, the blister wetlands of Kenya could be considered as groundwater discharge pools (Owen et al., 2004) where groundwater discharge that may be controlled by geology or topography, depending whether the groundwater has encountered a barrier or not. Even some Arctic lakes, formed in shallow topographic depressions, receiving groundwater input and seasonally situated within a stream

of snowmelt runoff (Gibson, 2002) can be considered topographically-controlled groundwater discharge pools. Within the humid landscape of Southeastern USA, Deemy and Rasmussen (2017)) also describe a vast number of pools along intermittent streams. These pools, which are seasonally connected by surface flows during the wet season, are expressions of the karst groundwater networks that underlie them and may be considered special cases of topographically-controlled groundwater discharge pools. In other humid regions, where rivers flow perennially,

pools within the river channel are perennially connected to upstream and downstream environments, and are rarely the subject of special investigation.

This framework is subject to refinement as sufficient data becomes available to fully characterise pool water balances and mode of occurrence. Snapshot data from multiple pools at one point in time can help distinguish perched pools vs groundwater discharge pools (i.e. pool water hydrochemically similar or different to rainfall or

groundwater), but in some cases water types are difficult to distinguish based on easily measured parameters like electrical conductivity or stable isotopes of water (Bourke et al., 2015). Highly instrumented sites with robust geological mapping, monitoring wells and temporal hydrologic data and are required to be confident of pool mode of occurrence. Pools can also be a mixture of types; through-flow pools with some regional groundwater input are likely common. The relative proportions of alluvial and regional groundwater in these pools will vary temporally

as seasonal recharge drains out of the alluvium, but quantification of these different water balance components is non-trivial.

We suggest that the development of empirical data sets that facilitate improved interpretation of snapshot data within the context of highly instrumented archetypal sites would be a beneficial next step. Extension of this framework to incorporate biological processes is also desirable. The nutrient and carbon transport between pools

during flows and the effects of anthropogenic disruption to groundwater inputs or surface water flushes into these pools is also not well known. These disruptions can be detrimental to water quality if the anthropogenic inputs are contaminated (Jackson 2010), but may also support seasonal connectivity that benefits the ecosystem by distributing nutrients and organic matter between pools (Jaeger et al., 2014). Effects of climate change (e.g. lower groundwater levels, thermal loading, and altered storm cycles) also combine with geomorphological and

biological factors to impact ecosystem function, but these mechanisms are not yet well understood.

The study of persistent river pools is a developing science and much remains to be done. Policy makers increasingly require accurate information on the mode of occurrence of surface water pools to put forward management planes to mitigate and/or minimise the adverse impacts of human activities (Leibowitz et al., 2008). Persistent pools exist in arid and semi-arid climates across the globe, and consistent data on geomorphology,

hydrology and ecology should be collected at each type of feature so that generalized patterns and processes can



be elucidated. We suggest that the process of understanding pool occurrence is an iterative one. That is to say, some data must be collected to infer the mechanism supporting the pool (e.g. geological mapping, water levels, salinity), but also an understanding of the pool mode of occurrence can be used to inform appropriate monitoring regimes. For example, pools that are supported by the discharge of deep regional groundwater are potentially vulnerable to groundwater abstraction, while perched pools are unlikely to be impacted. Thus, if managing impacts from groundwater abstraction then monitoring efforts would be best directed to the groundwater-dependant pools at the expense of pools that are disconnected from the groundwater system. A robust understanding of pool mode of occurrence requires a substantial investment of resources and is best supported by time-series data. Given limited resources, we suggest that time series data from fewer pools is more likely to provide more useful insights than snapshot data from many pools, particularly if pools can be grouped by landscape position or geology based on the framework presented here.

## 7 Conclusion

Most of the semi-arid regions of the globe are currently subject to increasing pressure from altered hydrology associated with anthropogenic activities such as water abstraction for agriculture and mining, as well shifting climates. These hydrological changes have the potential to impact persistent pools along river channels, which have ecological and social value. The study of these pools is a developing field, and a more robust framework for understanding their occurrence is required to support water management decisions. This paper summarizes the key hydrological processes and features of persistent pools, and identifies four dominant mechanisms that support pool persistence. Each mechanism has varying degrees of connection to groundwater and differing controls on groundwater sources (geological barrier vs topography). Susceptibility to hydrological change depends on the mechanism of pool persistence and the spatial distribution of stressors relative to the pool. While it has been developed based on a subset of pools within the Hammersley Basin we believe this framework is broadly applicable to persistent pools around the world. Our intention is for this framework to facilitate discussions based on common language and understanding so that risks to water quality or quantity (or a combination of both) in persistent pools can be adequately addressed.

## Acknowledgements

This paper is based on data collection funded by Rio Tinto Iron Ore and funding from the Australian Research Council, grant LP120100310. Author Shanafield's contribution was supported by funding from the Australian Research Council, grant DE150100302.





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

**Table 1 Summary of hydrogeomorphological framework for persistent pools**

| Pool setting and water balance* | Physical characteristics | Hydrochemical characteristics | Susceptibility to stressors |
|---|---|---|---|
| **Perched pools** $$\frac{\partial V}{\partial t} = EA$$ | Topographic low that catches rainfall/runoff. Present in i) elevated hard-rock headwaters of catchments and ii) regionally low-lying topographic location. Water levels in aquifer lower than pool water levels. Vertical head gradient between pool and aquifer with unsaturated zone below pool. | Highly variable; hydrochemistry is a function of rainfall and subsequent evaporation. Substantial enrichment of solutes and water isotopes during dry season. Precipitates are salts that wash away in next flood, or do not form because it's rainfall not groundwater. | Relies on surface flows and overland runoff, which is directly tied to precipitation. Sensitive to climate but largely independent of groundwater use. Where infiltration capacity is high pools in downstream areas are more vulnerable to reduced rainfall. |
| **Through-flow pools** $$\frac{\partial V}{\partial t} = Q_i - Q_o - EA$$ | Expression of river alluvium water table and through-flow. Head gradient reflects water table in alluvium. Water levels in pool coincident with water level in adjacent alluvium (cm-scale gradients expected at influent or effluent zones). Bank storage is important for pool water balance. Absence of surface geological features (e.g. hard-rock ridges) or waterfalls. Physical location may migrate as flood-scour re-shapes alluvium bedform. | Hydrochemically close to rainfall directly after flood, followed by enrichment of Cl and water isotopes during dry season, limited by through-flow. Flood water flushes through the alluvium and replaces or mixes with any residual stored water (i.e. hydrochemically flood and alluvial groundwater are the same after a flood). More through-flow means shorter pool residence time and less enrichment. | Relatively small changes in rainfall or groundwater level can result in pool drying if the water level in the unconfined (alluvial) aquifer is reduced to below the base of the pool. Impact of groundwater abstraction depends on volume and proximity to pool. Abstraction from regional aquifers that are hydraulically connected to alluvium may also effect pool water levels by inducing downward leakage from alluvium. |
| **Groundwater discharge pools** | | | |
| 1) Geological contacts and barriers to flow $$\frac{\partial V}{\partial t} = Q_i - Q_o - EA$$ | Two sub-types: i) Catchment constriction across ridges, or ii) aquifer thinning due to geological barrier intersecting topography. Presence of waterfalls or surface geological features (hard-rock ridges). Hydraulic head step-changes across pool feature. Carbonate deposits if source aquifer has sufficient alkalinity. | Consistent hydrochemical composition at point of contact/barrier. Evapo-concentration and evaporative enrichment down-gradient of discharge point. During rain event initial pulse of saline water due to runoff, followed by pool water completely replaced by flood water with fresh EC, then groundwater seeps in over dry season and equilibrates back to aquifer chemistry. | Susceptibility to groundwater abstraction depends on scale of source groundwater reservoir (if large then potentially more resilient) and location of groundwater abstraction. Water persistence is less susceptible to changes in rainfall than other pool types. Presence of geological barrier between pool and groundwater abstraction may limit impacts. |
| 2) Topographically controlled seepage from regional aquifer $$\frac{\partial V}{\partial t} = Q_i - EA$$ | Topography intersects i) water table or ii) preferential flow from artesian aquifer. Standing water persists during dry season due to groundwater discharge in absence of rainfall. For ii) no recharge to confined aquifer during flood event (pool is regional discharge zone). Carbonate deposits if source aquifer has sufficient alkalinity. | Consistent hydrochemical composition at point of seepage. For type i) during rain event initial pulse of saline water due to runoff, followed by pool water completely replaced by flood water with fresh EC, then groundwater seeps in over dry season and it equilibrates back to aquifer chemistry. | Susceptibility to groundwater abstraction depends on scale of source groundwater reservoir (if large then potentially more resilient) and location of groundwater abstraction. Hydraulic-gradient supporting pools may be similar to pool depth. No geological barrier to limit susceptibility. |

*Water balance of residual pool when disconnected from surface water flows





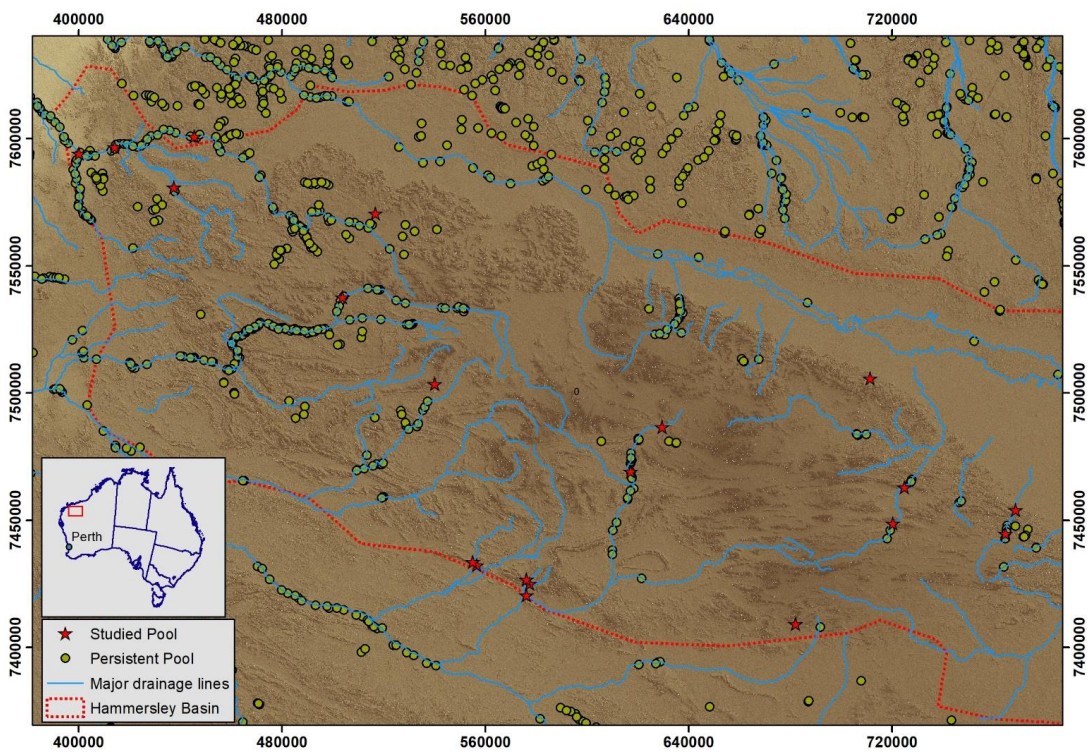

**Figure 1 Prevalence of persistent pools on watercourses in the Hammersley Basin ("Waterholes" features from Geodata Topo 250K Series 3 data set, http://pid.geoscience.gov.au/dataset/ga/63999) and select pools examined in detail for this study.**

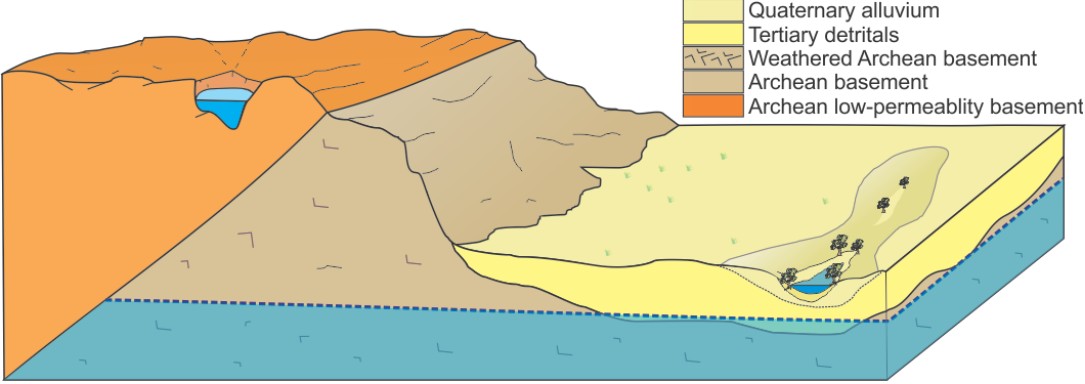

**Figure 2 Schematic illustration of a perched pool where rainfall-runoff collects in a depression that has morphology that limits evaporation, allowing water to be retained for an extended duration.**




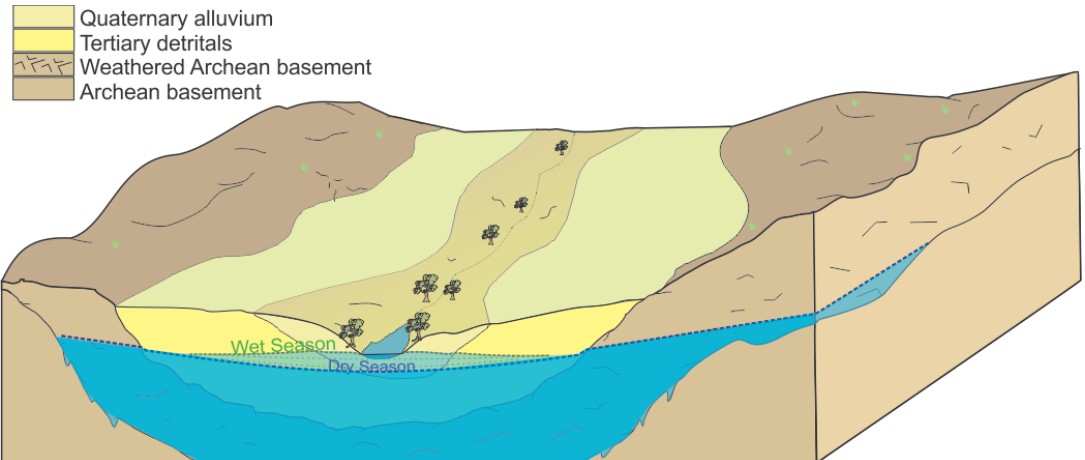

**Figure 3 Schematic illustration of a throughflow pool which are supported by bank storage inflow from the adjacent alluvial sediments.**

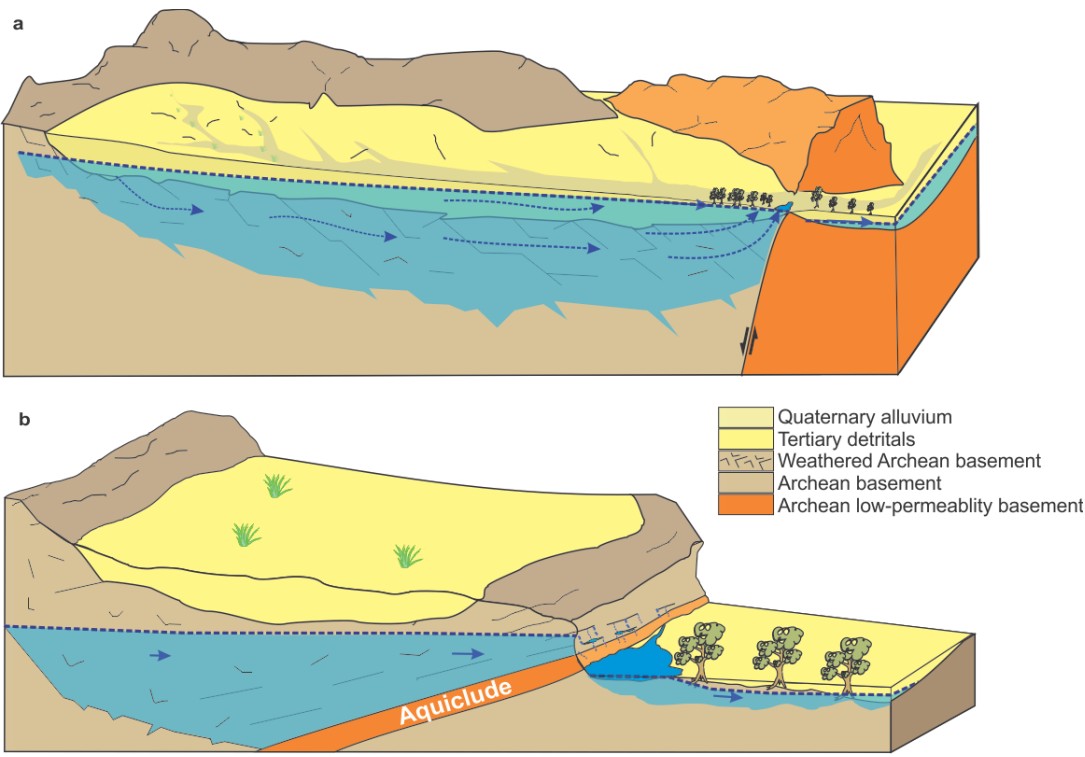

**Figure 4 Schematic illustration of a groundwater discharge pools where surface water persistence is driven by geological barriers that a) form a catchment constraint, or b) cause an aquifer to pinch out**

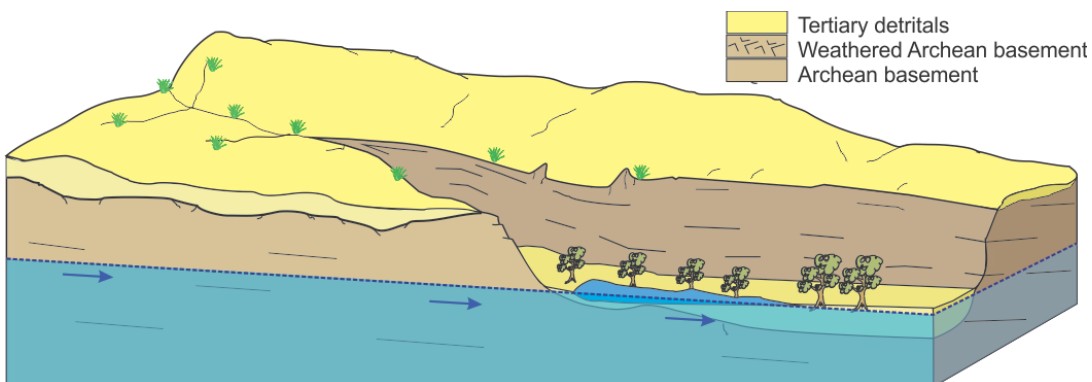

**Figure 5 Schematic illustration of a topographically-controlled pool receiving groundwater outflow from an unconfined regional aquifer.**

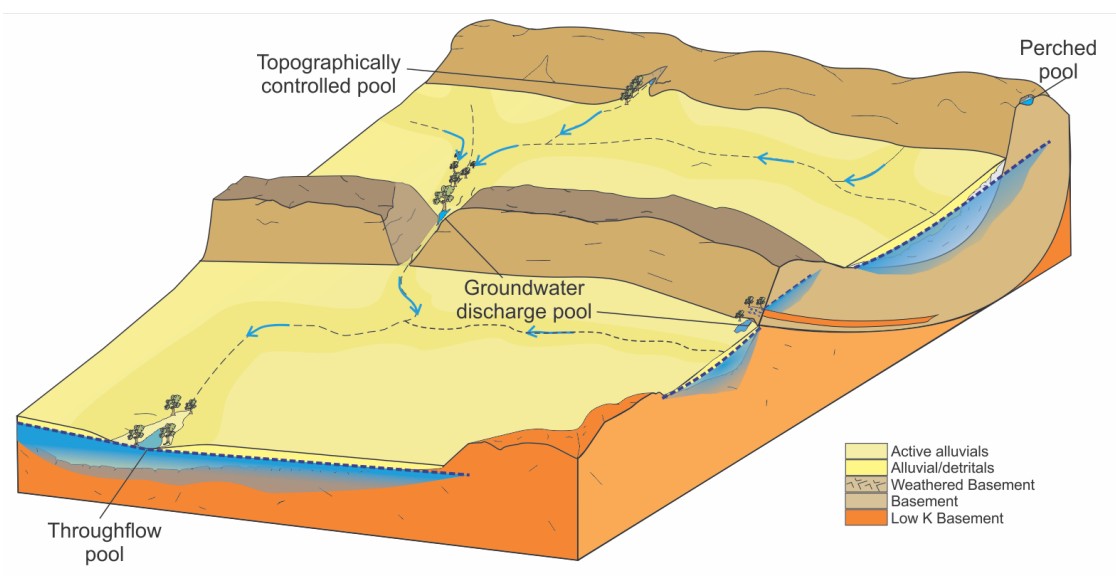

**Figure 6 Generalized landscape position of each type of persistent pool.**





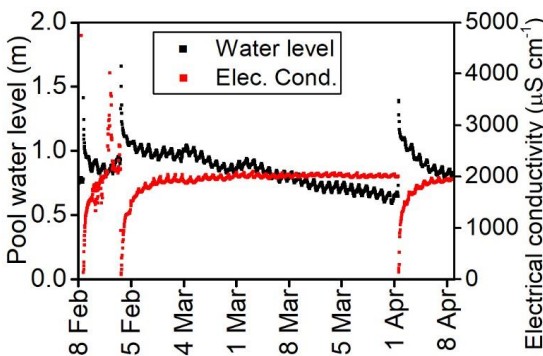

**Figure 7 EC time-series showing saline pulse and re-equilibration during dry season (data from a geologically controlled groundwater discharge pool).**

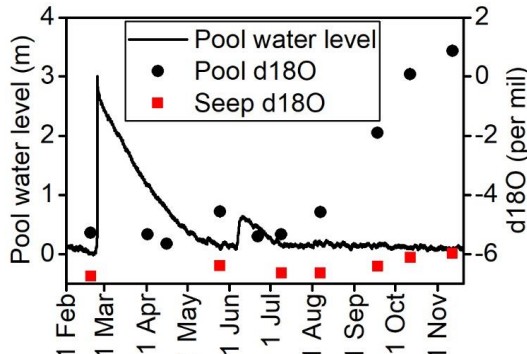

**Figure 8 Water levels and d18O showing stable isotopic composition at seep and evaporative enrichment down-gradient of seep (pool supported by groundwater discharge at geological contact).**

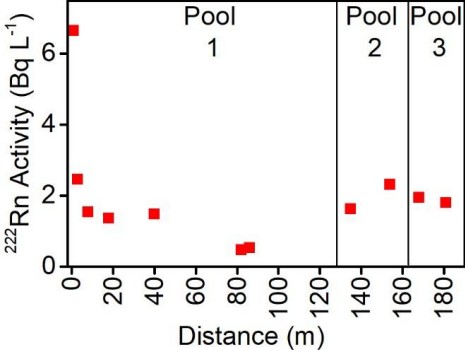

**Figure 9 Radon activities along a sequence of persistent pools showing spatial variation within 100's of metres (pool supported by a combination of through-flow and groundwater discharge from outcrop of regional aquifer).**