# Peer review of "A hydrological framework for persistent river pools in semiarid environments"

_Hydrology and Earth System Sciences, 2020_

## Short Comment (SC1) · 27 Apr 2020

This discussion manuscript is a much needed contribution to the relatively small but slowly growing body of literature about dryland surface water-groundwater interactions. We wish to make a few recommendations that could further improve this contribution.

Section 5 (Susceptibility of persistent pools to changing hydrological regimes) considers changes to persistent river pools over time. It is well known that streams located in the most arid regions of the world receive runoff and can show significant streamflow after rainfall events. The dynamics of these are complicated but have been discussed, for example in Acworth et al. (2016). The hydrological regimes for through-flow pools following surface water flows were unraveled in detail in Rau et al. (2017). For an ex-

ample please refer to their Figure 10. The sequence of events is generic for streams in unconsolidated sediments and some of the regimes would also apply to other types of pools. It would significantly improve this section with a better integration and discussion of the existing literature.

While the majority of persistent pools have their evapotranspirative water loss buffered by groundwater inflow, a better connection should be made to the existing dryland groundwater resource literature. For example groundwater recharge, surface water-groundwater interactions and environmental tracers in this context. Here, it would be beneficial to cite some key literature beyond what the authors have contributed (e.g., Scanlon et al., 2006; Herczeg and Leaney, 2011; Winter et al., 1998).

Great to see the concept of groundwater response times included. Our recent work has shown that longer response times are often associated with increased aridity (Cuthbert et al., 2019) so that may be worth adding in support of your arguments. Notably, that may also be the reason why groundwater fed pools were such an important resilient resource under fluctuating climates in the past in semi-arid areas during key periods for human evolution and dispersal (Cuthbert et al., 2017). The paleo-importance of persistent pools was likely very profound in many parts of the world so might be nice to add some discussion on this aspect as one more additional reason why such sites are important to understand.

The schematic cross section figures illustrate the paper really nicely.

Review comments were written by Gabriel C. Rau, Mark O. Cuthbert and Martin S. Andersen.

Recommended literature:

Acworth RI, Rau GC, Cuthbert MO, et al (2016) Long-term spatio-temporal precipitation variability in arid-zone Australia and implications for groundwater recharge. Hydrogeol J 24:905–921. doi: 10.1007/s10040-015-1358-7

Cuthbert, M. O. et al. (2019). Global patterns and dynamics of climate-groundwater interactions. Nature Climate Change, 9, 137-141. DOI: 10.1038/s41558-018-0386-4

Cuthbert, M. O., et al. (2017). The role of groundwater hydro-refugia in hominin evolution and dispersal. Nature Communications 8, 15696, doi:10.1038/ncomms15696

Herczeg AL, Leaney FW (2011) Review: Environmental tracers in arid-zone hydrology. Hydrogeol J 19:17–29. doi: 10.1007/s10040-010-0652-7

Rau GC, Halloran LJS, Cuthbert MO, et al (2017) Characterising the dynamics of surface water-groundwater interactions in intermittent and ephemeral streams using streambed thermal signatures. Adv Water Resour 107:354–369. doi: 10.1016/j.advwatres.2017.07.005

Scanlon BR, Keese KE, Flint AL, et al (2006) Global synthesis of groundwater recharge in semiarid and arid regions. Hydrol Process 20:3335–3370. doi: 10.1002/hyp.6335

Winter TC, Harvey JW, Franke OL, Alley WM (1998) Ground Water and Surface Water - A Single Resource. US Geological Survey, Circular 1139.
* * *

---

## Referee Comment (RC1) · Anonymous Referee #1 · 9 Jun 2020

The manuscript "A hydrological framework for persistent river pools in semi-arid environments" by Bourke, Shanafield, Hedley, and Dogramaci, attempts to present a framework of four hydrogeological categories for "persistent river pools" supposedly based on the study of 22 pools in otherwise ephemeral or intermittent streams in the Hamersley Basin of NW Western Australia. The manuscript states that of the numerous pools in this area (attested to by a Geodata Topo 250K Series 3 data set as "waterholes" shown in their Fig. 1), 22 were examined in detail. However, almost no data is presented pertaining to this supposed investigation of the 22 pools. Three poor quality data graphs are tacked on at the end of the paper, appearing as an after-thought, and are discussed very briefly in two paragraphs. The reader assumes that this "data" is from one or a few of the 22 pools supposedly studied. The bulk of the manuscript can be described

as introductory and speculative. Except for these three graphs (in Figures 7, 8, and 9) and summary descriptions of the four different categories in Table 1, no other "data" is presented. Thus, there is very little content presented in this manuscript. For these reasons, this manuscript is disappointing and poor scholarship. As is, the manuscript attempts to make the case for a hydrogeological division of persistent streams in arid-semi-arid areas, but provide no case studies. This manuscript cannot be considered a review paper of previous work on such pools because it does not come anywhere close to presenting that kind of literature survey. What the manuscript does do at length in the discussion is to advise and suggest ways to make progress in the area of identifying persistent pools in the system presented. The manuscript has laid out a case for the division and done the introductory work, however, as already stated in this review, there is almost no actual data on actual pools. What is needed are details of the pools said to be studied in detail with hydrogeological cross-sections and environmental tracer data which sets out the argument as to why the pool categorization presented is sound.

In my opinion HESS should not have sent this manuscript out for review. It should have been rejected by the editor/associate editors. I reject it without equivocation.

In summary, if there were a series of well documented case studies of these "persistent pools" with hydrogeological and environmental tracer data from the Hamersley area from which a hydrogeological framework can be based on, then this material should have been the focus of the manuscript. Finally, I note that there is a 2012 Journal of Hydrology paper by one of the co-authors (Dogramaci et al., 2012, J. of Hydrology 475, 281-293) which has environmental tracer data from many of the pools depicted in Fig. 1 (from the Bourke et al. manuscript). However, this data is not discussed or mentioned nor is this paper referenced. Again, poor scholarship.

---

## Author Comment (AC1) · 23 Jun 2020

Thank you for your comments on the manuscript. With this note we wish to briefly clarify the basis for the framework and the focus of the paper. We look forward to responding to your comments in full when we submit a revised manuscript on completion of this discussion phase. The framework proposed was developed by applying well-established principles in groundwater – surface water interaction and spring occurrence to the question of persistent pools along intermittent rivers. These concepts are robust and well accepted by the scientific community (see comments by Rau et al within this discussion). We acknowledge that further citation and reference to established literature would be beneficial and will gladly do so in a revised manuscript. The framework is also consistent with data that has been collected during field investigations at 22 pools in north-western Australia. We intentionally focused the paper on the conceptual framework, which is a summary of learnings from these investigations. Site-specific was kept to a minimum so as to keep the manuscript from being too long (there are 9 Figures in the submitted manuscript). We are certainly able to include more site-specific data if that is beneficial to the reader and can do so in our revised manuscript.

———————————————

---

## Referee Comment (RC2) · Anonymous Referee #2 · 6 Jul 2020

In "A hydrological framework for persistent river pools in semi-arid environments," Bourke et al. provide a framework for classifying various types of persistent pools encountered in the Hammersly Basin (Western Australia), and then suggest diagnostic tools that may be used to investigate and describe these different systems. Potential implications of a changing hydrologic regime are presented, and the discussion focuses largely on the next steps in the study of these pools.

I struggled to find novelty in this manuscript. Aside from perched pools, the persistent pools described by the authors are groundwater fed features, thus springs. The authors cite several studies that describe and/or classify springs based on (among other criteria) geomorphic features, and all the persistent pools they describe can fit into one existing category or another (n.b.: I would say that the type they claim has not been described in the literature is a limnocrene as described by Springer et al. 2009). Instead of relating their descriptions to one or another of these systems, the authors present a new classification framework without providing a compelling reason to abandon existing classification systems.

The diagnostic tools presented are wide ranging, and the fact that the manuscript include methods that are not often used together is a positive. The drawback to this presentation, though, is the lack of either data and rigorous analysis or extensive literature review on the use and limitations of those methods. The manuscript describes in a qualitative way different physical processes and posits how those processes would affect data collected by the different methods. Those descriptions are certainly reasonable and are largely intuitive, but they lack details and are unsupported by data from the study area. Almost no data are presented; the data that are presented are not rigorously analyzed but instead are used to superficially illustrate one of the posited phenomena. The discussion of climate change is similar – qualitative changes are presented with no real quantitative data about what has changed to date or expectations for future changes.

In the discussion section, the authors certainly note that there is significant work to do and suggest several next steps that are both logical and important. The results of those next steps would contribute to the literature in a way that this manuscript does not. Overall, the manuscript seems more like the introduction to a larger paper or to a proposal than it does a work that can stand on its own.

---

## Author Comment (AC2) · 7 Jul 2020

Thank you to the reviewer for taking the time to review our paper and give useful and constructive comments. We appreciate the feedback and constructive criticism, which has identified important aspects to be improved and will help us to refine and clarify the value and contribution of the paper. The major criticism of this reviewer is that classification systems for springs exist; almost all persistent pools are springs; therefore this paper is redundant. The reviewer is indeed correct that multiple classifications for springs do exist and we have referred to and cited some of these. However we disagree that the literature on springs forms a suitable and comprehensive framework for the study of persistent river pools. It is our experience that persistent river pools display hydrological characteristics that span streams and lakes as well as groundwater springs. Streams and lakes have been treated distinctly to groundwater springs within existing literature with concepts around gaining and losing streams, through-flow lakes, and hyporheic exchange have been well documented by experts in surface water – groundwater interaction (see Winter 1998 USGS Circular 1139 for a summary). Separately, there is an established body of work around mechanisms for spring discharge with multiple classifications to choose from. Spring classifications can be based on geological mechanism, hydrochemical properties, landscape setting, or a combination of all three, leading to broad categories such as thermal or artesian, as well as nuanced distinctions based on detailed geological structures (Alfaro 1994). For the purposes of understanding persistent river pools, this array of categories is both overly complex and incomplete.

To demonstrate, we examine the reviewers' suggestion that one of the pool types in this paper is a limnocrene spring after Springer (2009). We assume in this comment the reviewer is referring to what we have called through-flow pools. Springer (2009) presents a classification of springs based on their "sphere of influence", which is the setting into which the groundwater flows. A "limnocrene spring" is simply any groundwater that discharges to a pool, as distinct from say a "cave spring", which emerges into a cave. On this basis, one might consider all persistent pools that are not perched as limnocrene springs. However, the schema also articulates 'helocrene springs" which are associated with wetlands and "rheocrene springs" that emerge into stream channels. These also seem to be potentially fitting labels for persistent river pools, which does one choose? As an aid for the reader Springer (2009) includes a number of conceptual diagrams of their generalized spring types, similar in intent to the cross-section diagrams presented in our manuscript. Looking at the diagram of a limnocrene (Fig 1j) the reader is presented with a pool that is fed by groundwater that sourced from an aquifer that is karstic limestone or fractured (the watertable has stepwise changes) adjacent to and beneath the unconsolidated alluvium that the pool sits within. The groundwater flows from the limestone up a fault under pressure into the pool, and the alluvium directly below the pool is represented as unsaturated. So there are two issues

with the using this category to describe our through-flow pools. Firstly, the diagram is hydraulically inaccurate; a low-permeability confining layer between the limestone and the pool would be required for the alluvium for the groundwater to be under sufficient pressure to maintain a pool while the alluvium remains unsaturated, but no such layer is shown. Secondly, this diagram does not represent the hydraulic characteristics of through-flow pools, which are windows into the water level in the saturated zone within the modern creek channel. If the alluvium were not saturated to the level of the base of the pool then these types of pools would not exist. So the diagram is both inaccurate and inappropriate for the types of pools we have identified as through-flow pools.

A more appropriate conceptualization for the hydrology of the pools we have called through-flow pools, is that of through-flow lakes, which are widely documented within the literature on surface water – groundwater interactions (e.g. Fig 16C in Winter 1998). The hydrology of these lakes provides an excellent analogue for persistent river pools that are fed by water from within the alluvium of the river channel. Unlike spring categorizations (which focus geological controls and the point of groundwater outflow), the key features of this conceptualisation are that that 1) the surface water is connected to the underlying aquifer and represents a window to the water table and 2) that water enters the lakes (or pool) on the up-gradient side (the capture zone) and leaves the lake (or pool) on the down-gradient side (the release zone) (Townley and Trefry, 2000). The lake thus provides something of a "short circuit" for regional groundwater flow. The hydraulic mechanism is effectively identical to our through-flow pools; the pool is connected to a highly conductive saturated zone within the alluvium and the water level reflects the water table within the alluvium. Water within the alluvium is flowing in the down-gradient direction, but some portion of this subsurface water will use the pool as a kind of "short-circuit" to make its way downstream.

A comprehensive understanding of the hydrological attributes of persistent river pools, and the mechanisms that support them, therefore requires drawing on concepts from both surface water groundwater interactions, and spring hydrogeology. While numer-

ous classifications of spring systems have been published, these are commonly complex classification systems with multiple types of springs (often ten or more) that are neither designed for, nor easily applied to persistent river pools in a useful manner. Furthermore, these classifications do not include the full suite of mechanisms supporting the persistence of river pools. Similarly, as this reviewer notes, the methods required to understand the hydrology of persistent river pools are drawn from multiple strands of the literature and are not routinely deployed in tandem during studies of flowing streams, permanent lakes or groundwater springs. As such, there is a need for a comprehensive framework specific to persistent river pools that is inclusive enough to incorporate the range of pools encountered, but also simple enough so as to be useful for hydrologists, ecologists and water managers alike. We believe that is what we have presented in this manuscript. This context has clearly not been clearly articulated sufficiently in the existing manuscript and we look forward to improving this aspect of the paper in a revised submission.

Having established the need for this paper, we readily accept that it would benefit from a more robust treatment of existing literature as recommended by all three commentators thus far. We believe that the generalized diagrams presented in the current manuscript are vital templates that provide the framework that others site-specific data can sit within (see also comments by Rau et al in support of these diagrams). However, we are willing and able to include more site-specific data in the revised manuscript as both reviewers have suggested. We do not propose to include full case studies of individual pools as this would make the manuscript too long, but we can certainly see the value of including water level data demonstrating perched conditions as opposed to a through-flow pool, for example.

Thanks again for taking the time to provide a constructive review, it is much appreciated and we look forward to updating the manuscript to address your concerns and criticisms.

References:

Alfaro, C., Wallace, M. Origin and classification of springs and historical review with current applications, Environmental Geology, 24(2) 112-124, 1994.

Springer, A. E., and Stevens, L. E.: Spheres of discharge of springs, Hydrogeology Journal, 17, 83, 2009.

Townley, L. R., and Trefry, M. G.: Surface water-groundwater interaction near shallow circular lakes: Flow geometry in three dimensions, Water Resources Research, 36, 935-948, 10.1029/1999wr900304, 2000.

Winter, T. C., Harvey, J.W., Franke, O.L., Alley, W.M., Ground water and surface water: a single resource. US Geological Survey Circular 1139, 79pp, 1998.
* * *

---

## Author Comment (AC3) · 13 Jul 2020

Thank you very much for taking the time to provide constructive feedback on this manuscript. Your comments are well taken and provide valuable avenues for improving the paper. We are particularly pleased that as researchers with experience in this area of hydrology you have understood our intention and see clear value in the paper. We will certainly incorporate your suggestions for improvements to the manuscript during the revision process.